# On an Optimal Quadrature Formula in a Hilbert Space of Periodic Functions

**Kholmat Shadimetov** [1,2], **Abdullo Hayotov** [2,3,*] and **Botir Abdikayimov** [1]

1  Department of Informatics and Computer Graphycs, Tashkent State Transport University, 1, Odilkhodjaev Str., Tashkent 100167, Uzbekistan
2  V. I. Romanovskiy Institute of Mathematics, Uzbekistan Academy of Sciences, 4b, University Str., Tashkent 100174, Uzbekistan
3  Department of Computational Mathematics and Information Systems, National University of Uzbekistan named after M.Ulugbek, 4, University Str., Tashkent 100174, Uzbekistan
*  Correspondence: hayotov@mail.ru; Tel.: +998-94-618-65-53

**Abstract:** The present work is devoted to the construction of optimal quadrature formulas for the approximate calculation of the integrals $\int_0^{2\pi} e^{i\omega x} \varphi(x) dx$ in the Sobolev space $\widetilde{H}_2^m$. Here, $\widetilde{H}_2^m$ is the Hilbert space of periodic and complex-valued functions whose $m$-th generalized derivatives are square-integrable. Here, firstly, in order to obtain an upper bound for the error of the quadrature formula, the norm of the error functional is calculated. For this, the extremal function of the considered quadrature formula is used. By minimizing the norm of the error functional with respect to the coefficients, an optimal quadrature formula is then obtained. Using the explicit form of the optimal coefficients, the norm of the error functional of the optimal quadrature formula is calculated. The convergence of the constructed optimal quadrature formula is investigated, and it is shown that the rate of convergence of the optimal quadrature formula is $O(h^m)$ for $|\omega| < N$ and $O(|\omega|^{-m})$ for $|\omega| \geq N$. Finally, we present numerical results of comparison for absolute errors of the optimal quadrature formula with the $\exp(i\omega x)$ weight in the case $m = 2$ and the Midpoint formula. There, one can see the advantage of the optimal quadrature formulas.

**Keywords:** Sobolev space of periodic functions; extremal function; error functional; optimal quadrature formulas; oscillating functions

**MSC:** 65D30; 65D32

## 1. Introduction and Statement of the Problem

Approximate calculation of the integral

$$\int_0^{2\pi} \exp(i\omega x)\varphi(x)dx, \tag{1}$$

where $\omega$ is an integer and $i^2 = -1$, plays an important role in computational mathematics. It is well-known that numerical calculation of such integrals encounters difficulties for large values of $\omega$ because the integrand oscillates strongly. Computation of the integral (1) are often done by the Filon method [1]. The Filon method reminds us of the Simpson quadrature formula. However, while in the Simpson method the entire integrand is replaced by a parabola, in the Filon method, only the function $\varphi(x)$ is replaced by a parabola. This way Filon obtained the quadrature formula with coefficients depending on $\omega$ [1].

Various aspects of the calculation of the integral (1) are discussed in a number of papers [2–16], where there are fairly complete bibliographies.

In the present work, combining S.L. Sobolev's and I. Babuška's methods for the approximate computation of the integral (1), an optimal quadrature formula is constructed,

and the square of the norm for the error functional of the obtained optimal quadrature formula is evaluated in a certain Hilbert space.

Suppose $\widetilde{H}_2^{(m)}$, $m \geq 1$ is the Hilbert space of $2\pi$-periodic, complex-valued functions $\varphi(x)$, $-\infty < x < \infty$, which are square-integrable with the $m$th generalized derivative. Every element of the space $\widetilde{H}_2^{(m)}$ is a class of such functions that differ from each other by a constant term, and the inner product of the elements in this space is defined as

$$(f, g)_m = \int_0^{2\pi} f^{(m)}(x) \overline{g^{(m)}(x)} dx. \tag{2}$$

Here, the notation $\bar{g}$ is the complex conjugate to $g$. The norm of a function $\varphi(x)$ in the space $\widetilde{H}_2^{(m)}$ is defined by

$$\left\| \varphi | \widetilde{H}_2^{(m)} \right\| = \left( \int_0^{2\pi} \varphi^{(m)}(x) \overline{\varphi^{(m)}(x)} \right)^{\frac{1}{2}}. \tag{3}$$

For $\varphi(x) \in \widetilde{H}_2^{(m)}$, we consider a quadrature formula of the form

$$\int_0^{2\pi} \exp(i\omega x) \varphi(x) dx \cong \sum_{k=1}^{N} C_k \varphi\left(\frac{2\pi k}{N}\right), \tag{4}$$

where $C_k$ are coefficients of the quadrature formula, $N = 2, 3, \ldots$.

The error of the quadrature formula is given in the form

$$\begin{aligned}
(\ell, \varphi) &= \int_0^{2\pi} \exp(i\omega x) \varphi(x) dx - \sum_{k=1}^{N} C_k \varphi\left(\frac{2\pi k}{N}\right) \\
&= \int_0^{2\pi} \left[ \exp(i\omega x) - \sum_{k=1}^{N} C_k \sum_{\beta=-\infty}^{\infty} \delta\left(x - \frac{2\pi k}{N} - 2\pi\beta\right) \right] \varphi(x) dx.
\end{aligned} \tag{5}$$

Here, $\delta(x)$ is the Dirac delta-function,

$$\ell(x) = \exp(i\omega x) - \sum_{k=1}^{N} C_k \sum_{\beta=-\infty}^{\infty} \delta\left(x - \frac{2\pi k}{N} - 2\pi\beta\right), \tag{6}$$

$\ell$ is the periodic functional of the error for the quadrature formula, and $(\ell, \varphi)$ is the value of the error functional at $\varphi$.

**The challenge** in the construction of an optimal quadrature formula for the approximate calculation of the integral (1) is calculating the quantity

$$\inf_{C_k} \sup_{\varphi \in \widetilde{H}_2^{(m)}} \frac{|(\ell, \varphi)|}{\left\| \varphi | \widetilde{H}_2^{(m)} \right\|} = \left\| \mathring{\ell} | \widetilde{H}_2^{(m)*} \right\| = \left\| \mathring{\ell} \right\|, \tag{7}$$

that is, finding a function $\psi_\ell(x)$ from $\widetilde{H}_2^{(m)}$ for which the exact upper bound is attained, and the coefficients $\mathring{C}_k$ for which the exact lower bound are attained in (7).

In this case, the function $\psi_\ell(x)$ is said to be *the extremal* for the given quadrature formula, $\mathring{C}_k$ values are called the *the optimal coefficients* for the quadrature formula (4) in the space $\widetilde{H}_2^{(m)}$, and $\|\mathring{\ell}\|$ is the norm for the error functional of the optimal quadrature formula.

It should be noted that $\widetilde{H}_2^{(m)*}$ is the conjugate space to the space $\widetilde{H}_2^{(m)}$, and it consists of all periodic functionals that are orthogonal to the unity, i.e.,

$$(\ell, 1) = 0. \tag{8}$$

The remaining part of the paper is organized as follows. In Section 2, we present the main results; i.e., we obtain the extremal function, the analytic expressions for the optimal coefficients, and the norm of the error functional for the optimal quadrature formula of the form (4). Further, in Sections 3–6, we provide proofs of the main results.

## 2. Main Results

The central results of the present paper are the following.

**Theorem 1.** *The extremal function of the periodic error functional $\ell$ for the quadrature formula (4) has the form*

$$\psi_\ell(x) = \frac{\exp(-i\omega x)}{\omega^{2m}} - \sum_{k=1}^{N} \overline{C}_k \sum_{\beta \neq 0} \frac{\exp\left(i\beta\left(x - \frac{2\pi k}{N}\right)\right)}{2\pi\beta^{2m}} + \nu. \tag{9}$$

*Here, $\overline{C}_k$ are complex conjugate to the coefficients $C_k$ of the quadrature formula (4), and $\nu$ is an unknown number.*

**Theorem 2.** *If $\varphi(x) \in \widetilde{H}_2^{(m)}$, then, for the optimal coefficients of the quadrature formula (4) with the error functional (6), the following formula holds*

$$\mathring{C}_k = \frac{2\pi}{N}\left(\frac{\sin\frac{\pi\omega}{N}}{\frac{\pi\omega}{N}}\right)^{2m} \frac{(2m-1)! \cdot \exp\left(\frac{2\pi i\omega k}{N}\right)}{2\sum_{n=0}^{m-2} a_n^{(2m-2)}\cos(2\pi(m-1-n)\frac{\omega}{N}) + a_{m-1}^{(2m-2)}}, \tag{10}$$

$$k = 1, 2, \ldots, N.$$

*In the last formula, the $a_n^{(2m-2)}$ values are the coefficients of the Euler–Frobenius polynomial $E_{2m-2}(x)$ of degree $(2m-2)$ and are defined as follows:*

$$a_n^{(2m-2)} = \sum_{j=0}^{n}(-1)^j\binom{2m}{j}(n+1-j)^{2m-1},$$

$$\binom{2m}{j} = \frac{(2m)!}{(2m-j)! \, j!}.$$

**Theorem 3.** *In the space $\widetilde{H}_2^{(m)}$, the square of the norm of the error functional $\ell(x)$ for the optimal quadrature formula of the form (4) has the following expression:*

$$\left\|\mathring{\ell}|\widetilde{H}_2^{(m)*}\right\|^2 = \frac{2\pi}{\omega^{2m}}\left[1 - \left(\frac{\sin\frac{\pi\omega}{N}}{\frac{\pi\omega}{N}}\right)^{2m}\frac{(2m-1)!}{2\sum_{n=0}^{m-2} a_n^{(2m-2)}\cos(m-1-n)\frac{\omega}{N} + a_{m-1}^{(2m-2)}}\right]. \tag{11}$$

Hence, in particular, we obtain the following.

**Remark 1.** *For $\omega = 0$, we have the well-known optimal quadrature formula, which is the following rectangular formula*

$$\int_0^{2\pi}\varphi(x)dx \cong \frac{2\pi}{N}\sum_{k=1}^{N}\varphi\left(\frac{2\pi k}{N}\right).$$

*Moreover, for the norm of the error functiona, we obtain*

$$\left\| \mathring{\ell} | \widetilde{H}_2^{(m)*} \right\|^2 = \frac{\left(\frac{2\pi}{N}\right)^{2m} |B_{2m}|}{(2m)!},$$

*where $B_{2m}$ is the Bernoulli number.*

**Remark 2.** *If $\omega$ is a multiple for the number $N$ of the nodes of the quadrature formula, i.e., $\omega = Np$, $p = \pm 1, \pm 2, \ldots$, then*

$$\mathring{C}_k = 0, \quad k = 1, 2, \ldots, N$$

*and*

$$\left\| \mathring{\ell} | \widetilde{H}_2^{(m)*} \right\|^2 = \frac{2\pi}{\omega^{2m}}.$$

**Remark 3.** *Calculations show that*

$$\sum_{n=0}^{2m-2} a_n^{(2m-2)} = (2m-1)!,$$

*where $a_n^{(2m-2)}$ values are the coefficients of the Euler–Frobenius polynomial of degree $(2m-2)$.*

## 3. The Extremal Function to the Error Functional for the Quadrature Formula (4)

In order to find an explicit form of the norm for the error functional $\ell(x)$, we use an extremal function of the given functional, i.e., a function $\psi_\ell(x)$ for which the following holds:

$$(\ell, \psi_\ell) = \left\| \ell | \widetilde{H}_2^{(m)*} \right\| \cdot \left\| \psi_\ell | \widetilde{H}_2^{(m)} \right\|. \tag{12}$$

**Proof of Theorem 1.** The idea of the proof is as follows. Using the formula (2) and applying the Riesz theorem for the error functional $\ell$, we find an explicit expression for $\ell$ by a new function $\psi_\ell$, which is an element of the space $\widetilde{H}_2^{(m)}$. It is easy to see that the function $\psi_\ell$ is related to the error functional $\ell$ by the following differential equation:

$$\overline{\psi_\ell^{(2m)}(x)} = (-1)^m \left( \exp(i\omega x) - \sum_{k=1}^{N} C_k \sum_{\beta=-\infty}^{\infty} \delta\left( x - \frac{2\pi k}{N} - 2\pi\beta \right) \right). \tag{13}$$

Indeed, by the definition of the inner product, we have

$$(\ell, \varphi) = \int_0^{2\pi} \overline{\psi_\ell^{(m)}(x)} \varphi^{(m)}(x) dx. \tag{14}$$

Integrating by parts the right-hand side of (14) and taking into account that a function $\varphi(x)$ is infinitely differentiable and finite, i.e., $\varphi(x) \in \mathring{C}^{(\infty)}$, we obtain

$$(\ell, \varphi) = (-1)^m \int_0^{2\pi} \overline{\frac{d^{2m}\psi_\ell(x)}{dx^{2m}}} \varphi(x) dx. \tag{15}$$

On the other hand, by definition of the error functional $\ell(x)$, we have

$$(\ell, \varphi) = \int_0^{2\pi} \left[ \exp(i\omega x) - \sum_{k=1}^{N} C_k \sum_{\beta=-\infty}^{\infty} \delta\left( x - \frac{2\pi k}{N} - 2\pi\beta \right) \right] \varphi(x) dx. \tag{16}$$

Placing (16) on the left-hand side of (15), we find (13). Since the space of the infinity differentiable and finite functions is dense in the space $\widetilde{H}_2^{(m)}$, i.e., any function $\varphi(x)$ from $\widetilde{H}_2^{(m)}$ can be approximated with arbitrary high accuracy by a sequence of the functions from the space $\mathring{C}^{(\infty)}$, it follows that a periodic solution of Equation (13) is the extremal function $\psi_\ell$ of the error functional $\ell$ for the quadrature formula (4) and $\psi_\ell \in \widetilde{H}_2^{(m)}$.

Further, we find a periodic solution of Equation (13). In order to solve this equation, we use the Fourier transforms. For this, we provide some useful formulas:

$$F[\varphi](p) = \int_{-\infty}^{\infty} \exp(2\pi ipx)\varphi(x)dx, \tag{17}$$

$$F^{-1}[\varphi](x) = \int_{-\infty}^{\infty} \exp(-2\pi ipx)\varphi(p)dp, \tag{18}$$

$$F^{-1}[F[\varphi]] = \varphi(x), \tag{19}$$

$$F[\varphi^{(n)}(x)] = (-2\pi ip)^n F[\varphi], \tag{20}$$

$$F[\phi_0(x)] = \phi_0(p), \quad \phi_0(x) = \sum_{\beta=-\infty}^{\infty} \delta(x - \beta). \tag{21}$$

Applying the Fourier transform to both sides of Equation (13), we obtain

$$F\left[\overline{\frac{d^{2m}\psi_\ell(x)}{dx^{2m}}}\right] = (-1)^m (F[\exp(i\omega x)] - F[T(x)]), \tag{22}$$

where

$$T(x) = \sum_{k=1}^{N} C_k \sum_{\beta} \delta\left(x - \frac{2\pi k}{N} - 2\pi\beta\right). \tag{23}$$

By virtue of the formula (20), we have

$$F\left[\overline{\frac{d^{2m}\psi_\ell(x)}{dx^{2m}}}\right] = (-2\pi ip)^{2m} F[\psi_\ell(x)]. \tag{24}$$

Now, using the definition of the Fourier transform, i.e., Formula (17), we directly obtain

$$F[\exp(i\omega x)] = \delta\left(p + \frac{\omega}{2\pi}\right). \tag{25}$$

Applying the Fourier transform defined by Formula (17) to both sides of (23), we find

$$
\begin{aligned}
F[T(x)] &= \sum_{k=1}^{N} C_k F\left[\delta\left(x - \frac{2\pi k}{N} - 2\pi\beta\right)\right] \\
&= \sum_{k=1}^{N} C_k \int_{-\infty}^{\infty} \exp(2\pi ipx) \sum_{\beta=-\infty}^{\infty} \delta\left(x - \frac{2\pi k}{N} - 2\pi\beta\right)dx.
\end{aligned} \tag{26}
$$

Using the known properties of the delta-function, we have

$$\delta\left(x - \frac{2\pi k}{N} - 2\pi\beta\right) = \frac{1}{2\pi}\delta\left(\frac{Nx - 2\pi k}{2\pi N} - \beta\right)$$

and introducing the notation $y = \frac{Nx - 2\pi k}{2\pi N}$ and using the formula (21), we rewrite Equality (26) in the form

$$
\begin{aligned}
F[T(x)] &= \sum_{k=1}^{N} C_k \int_{-\infty}^{\infty} \exp\left(2\pi i p \left(2\pi y + \frac{2\pi k}{N}\right)\right) \sum_{\beta=-\infty}^{\infty} \delta(y - \beta) dy \\
&= \sum_{k=1}^{N} C_k \exp\left(2\pi i p \frac{2\pi k}{N}\right) \int_{-\infty}^{\infty} \exp(2\pi i (2\pi p) y) \sum_{\beta=-\infty}^{\infty} \delta(y - \beta) dy \\
&= \sum_{k=1}^{N} C_k \exp\left(2\pi i p \frac{2\pi k}{N}\right) \sum_{\beta=-\infty}^{\infty} \delta(2\pi p - \beta).
\end{aligned}
\tag{27}
$$

Now, from (22), (24), (25) and (27), it immediately follows that

$$
\begin{aligned}
&(-2\pi i p)^{2m} F[\overline{\psi}_\ell(x)] \\
&= (-1)^m \left[\delta\left(p + \frac{\omega}{2\pi}\right) - \sum_{k=1}^{N} C_k \exp\left(2\pi i p \frac{2\pi k}{N}\right) \sum_{\beta=-\infty}^{\infty} \delta(2\pi p - \beta)\right].
\end{aligned}
\tag{28}
$$

The right-hand side of (28) is equal to 0 at $p = 0$, since, in virtue of (8), the singularities

$$
\delta\left(p + \frac{\omega}{2\pi}\right) \text{ and } \sum_{k=1}^{N} C_k \exp\left(2\pi i p \frac{2\pi k}{N}\right) \sum_{\beta=-\infty}^{\infty} \delta(2\pi p - \beta)
$$

are mutually canceled.

Therefore, we can divide both sides of Equation (28) by $(-2\pi i p)^{2m}$. This division is not uniquely defined. For Equation (28), the function $F[\overline{\psi}_\ell]$ is defined up to the term of the form $\nu\delta(p)$. Taking into account the aforementioned as well as the properties of the delta-function, we obtain

$$
\begin{aligned}
F[\overline{\psi}_\ell] &= \frac{\delta\left(p + \frac{\omega}{2\pi}\right)}{(2\pi p)^{2m}} - \sum_{k=1}^{N} C_k \exp\left(\frac{(2\pi)^2 i p k}{N}\right) \sum_{\beta \neq 0} \frac{\delta(2\pi p - \beta)}{(2\pi p)^{2m}} \\
&= \frac{\delta\left(p + \frac{\omega}{2\pi}\right)}{\omega^{2m}} - \sum_{k=1}^{N} C_k \sum_{\beta \neq 0} \exp\left(\frac{2\pi i \beta k}{N}\right) \frac{\delta\left(p - \frac{\beta}{2\pi}\right)}{2\pi \beta^{2m}} + \nu\delta(p).
\end{aligned}
\tag{29}
$$

Applying the inverse Fourier transform to both sides of Equation (29) and using Formulas (18) and (19), after some calculations, we have

$$
\overline{\psi}_\ell(x) = \frac{\exp(i\omega x)}{\omega^{2m}} - \sum_{k=1}^{N} C_k \sum_{\beta \neq 0} \frac{\exp\left(i\beta\left(\frac{2\pi k}{N} - x\right)\right)}{2\pi \beta^{2m}} + \nu.
\tag{30}
$$

From (30), it follows that

$$
\psi_\ell(x) = \frac{\exp(-i\omega x)}{\omega^{2m}} - \sum_{k=1}^{N} \overline{C}_k \sum_{\beta \neq 0} \frac{\exp\left(i\beta\left(\frac{2\pi k}{N} - x\right)\right)}{(2\pi \beta)^{2m}} + \nu.
\tag{31}
$$

Theorem 1 is completely proved. □

Now, we give an important theorem on zeros of the extremal function which is due to I. Babuška. This theorem was proved in the language of functional analysis (see, for example [17,18]).

**Theorem 4** (I. Babuška). *Suppose the error functional* $(\ell, \varphi)$
*(a) is defined on the space* $\widetilde{H}_2^{(m)}$—*i.e., its value at constant is zero—and*
*(b) is optimal; i.e., among all functionals of the form*

$$\ell(x) = \exp(\mathrm{i}\omega x) - \sum_{k=1}^{N} C_k \sum_{\beta=-\infty}^{\infty} \delta\left(x - \frac{2\pi k}{N} - 2\pi\beta\right)$$

*with given system of nodes, it has the lowest norm in* $\widetilde{H}_2^{(m)*}$.
*Then there exists a solution* $\psi_\ell(x)$ *of the equation*

$$\overline{\frac{d^{2m}\psi_\ell(x)}{dx^{2m}}} = (-1)^m \ell(x),$$

*which is zero at points* $\frac{2\pi k}{N}$ *and belongs to* $\widetilde{H}_2^{(m)}$.

## 4. The Square of the Error Functional of the Quadrature Formula (4)

It is easy to proof Babuška's theorem also by algebraic way. For the sake of completeness, we give this proof here as well. A quadrature formula with the error functional $\ell(x)$ in the space $\widetilde{H}_2^{(m)}$ can be characterized by two manners. From one side, this quadrature formula is defined by coefficients $C_k$, $k, = 1, 2, \ldots, N$, under the condition

$$(\ell, 1) = 0. \tag{32}$$

From the other side, it is defined by the extremal function $\psi_\ell(x)$ of the quadrature formula. The square of the norm for the error functional of the quadrature formula is expressed by the bilinear form with respect to coefficients of the formula and values of the extremal function. Indeed, since $\widetilde{H}_2^{(m)}$ is a Hilbert space, we have

$$\left\| \ell | \widetilde{H}_2^{(m)*} \right\|^2 = (\ell, \psi_\ell) = \int_0^{2\pi} \psi_\ell^{(m)}(x) \overline{\psi_\ell^{(m)}(x)} dx = (-1)^m \int_0^{2\pi} \overline{\psi_\ell^{(2m)}(x)} \psi_\ell(x) dx,$$

where $\psi_\ell(x)$ is the extremal function of our quadrature formula. However,

$$\overline{\frac{d^{2m}\psi_\ell(x)}{dx^{2m}}} = (-1)^m \ell(x),$$

, so

$$\left\| \ell | \widetilde{H}_2^{(m)*} \right\|^2 = (\ell, \psi_\ell) = \int_0^{2\pi} \ell(x)\psi_\ell(x) dx. \tag{33}$$

Using Formulas (6), (9), and (33), we have

$$\left\| \ell | \widetilde{H}_2^{(m)*} \right\|^2 = \int_0^{2\pi} \left( \exp(\mathrm{i}\omega x) - \sum_{k=1}^{N} C_k \sum_{\beta=-\infty}^{\infty} \delta\left(x - \frac{2\pi k}{N} - 2\pi\beta\right) \right)$$

$$\times \left( \frac{\exp(-\mathrm{i}\omega x)}{\omega^{2m}} - \sum_{k'=1}^{N} \overline{C}_{k'} \sum_{\beta \neq 0} \frac{\exp\left(\mathrm{i}\beta\left(x - \frac{2\pi k'}{N}\right)\right)}{2\pi\beta^{2m}} + \nu \right) dx.$$

However, by virtue of (8), i.e., by equality $\sum_{k=1}^{N} C_k = 0$, we obtain

$$
\begin{aligned}
\left\| \ell | \widetilde{H}_2^{(m)*} \right\|^2 &= \int_0^{2\pi} \left( \exp(i\omega x) - \sum_{k=1}^{N} C_k \sum_{\beta=-\infty}^{\infty} \delta\left( x - \frac{2\pi k}{N} - 2\pi\beta \right) \right) \\
&\times \left( \frac{\exp(-i\omega x)}{\omega^{2m}} - \sum_{k'=1}^{N} \overline{C}_{k'} \sum_{\beta \neq 0} \frac{\exp\left( i\beta \left( x - \frac{2\pi k'}{N} \right) \right)}{2\pi\beta^{2m}} \right) dx \\
&= \frac{1}{\omega^{2m}} \int_0^{2\pi} \exp(i\omega x)\exp(-i\omega x)dx - \sum_{k'=1}^{N} \overline{C}_{k'} \sum_{\beta \neq 0} \frac{1}{2\pi\beta^{2m}} \\
&\times \int_0^{2\pi} \exp(i\omega x)\exp\left( i\beta \left( x - \frac{2\pi k'}{N} \right) \right) dx - \frac{1}{\omega^{2m}} \sum_{k=1}^{N} C_k \\
&\times \exp\left( \frac{-2\pi i\omega k}{N} \right) + \sum_{k=1}^{N} \sum_{k'=1}^{N} C_k \overline{C}_{k'} \sum_{\beta \neq 0} \frac{\exp\left( \frac{2\pi i\beta}{N}(k-k') \right)}{2\pi\beta^{2m}}.
\end{aligned}
$$

Simple calculations show that

$$
\frac{1}{\omega^{2m}} \int_0^{2\pi} (\exp(i\omega x)\exp(-i\omega x))dx = \frac{2\pi}{\omega^{2m}}. \tag{34}
$$

By virtue of

$$
\int_0^{2\pi} \exp(i(\omega+\beta)x)dx = \begin{cases} 0, & \text{if } \omega + \beta \neq 0, \\ 2\pi, & \text{if } \omega + \beta = 0 \end{cases}
$$

it follows that

$$
\begin{aligned}
&\sum_{\beta \neq 0} \frac{1}{2\pi\beta^{2m}} \int_0^{2\pi} \exp(i\omega x)\exp(i\beta(x - \frac{2\pi k'}{N}))dx \\
&= \sum_{\beta \neq 0} \frac{\exp\left( \frac{-2\pi k' i\beta}{N} \right)}{2\pi\beta^{2m}} \int_0^{2\pi} \exp(i(\omega+\beta)x)dx = \frac{2\pi}{\omega^{2m}} \exp\left( \frac{2\pi i\omega k'}{N} \right). \tag{35}
\end{aligned}
$$

Using Equalities (34) and (35) for the square of the norm for the error functional of the quadrature formula, we obtain the following analytical expression

$$
\begin{aligned}
\left\| \ell | \widetilde{H}_2^{(m)*} \right\|^2 &= \frac{2\pi}{\omega^{2m}} - \frac{1}{\omega^{2m}} \sum_{k'=1}^{N} \overline{C}_{k'} \exp\left( \frac{2\pi i\omega k'}{N} \right) - \frac{1}{\omega^{2m}} \sum_{k=1}^{N} C_k \exp\left( \frac{-2\pi i\omega k}{N} \right) \\
&+ \sum_{k=1}^{N} \sum_{k'=1}^{N} C_k \overline{C}_{k'} \sum_{\beta \neq 0} \frac{\exp\left( \frac{2\pi i\beta(k-k')}{N} \right)}{2\pi\beta^{2m}}. \tag{36}
\end{aligned}
$$

For finding the minimum of the square for the error functional of the quadrature formula, we apply the method of indefinite Lagrange multipliers. For this we consider the following function:

$$\Lambda(C, \overline{C}, \nu) = \left\| \ell | \widetilde{H}_2^{(m)*} \right\|^2 - \nu(\ell, 1).$$

Setting to 0 all partial derivatives by $C_k$, $\overline{C}_k$, and $\nu$ of the function $\Lambda(C, \overline{C}, \nu)$, we have

$$\frac{\partial \Lambda}{\partial C_k} = 0, \ k = 1, 2, \ldots, N,$$

$$\frac{\partial \Lambda}{\partial \overline{C}_{k'}} = 0, \ k = 1, 2, \ldots, N,$$

$$\frac{\partial \Lambda}{\partial \nu} = 0.$$

These give the following system of equations

$$\frac{\partial \Lambda}{\partial C_k} = -\frac{1}{\omega^{2m}} \exp\left(-\frac{2\pi \mathrm{i} \omega k}{N}\right) + \sum_{k'=1}^{N} \overline{C}_{k'} \sum_{\beta \neq 0} \exp\left(\frac{2\pi \beta \mathrm{i}(k - k')}{N}\right) - \nu = 0$$

$$\text{for } k = 1, 2, \ldots, N, \tag{37}$$

$$\frac{\partial \Lambda}{\partial \overline{C}_{k'}} = -\frac{1}{\omega^{2m}} \exp\left(\frac{2\pi \mathrm{i} \omega k'}{N}\right) + \sum_{k=1}^{N} C_k \sum_{\beta \neq 0} \frac{\exp\left(\frac{2\pi \beta \mathrm{i}(k - k')}{N}\right)}{2\pi \beta^{2m}} = 0,$$

$$\text{for } k' = 1, 2, \ldots, N, \tag{38}$$

$$\frac{\partial \Lambda}{\partial \nu} = \sum_{k=1}^{N} C_k = 0. \tag{39}$$

It is not difficult to see that

$$\frac{\overline{\partial \Lambda}}{\partial \overline{C}_{k'}} = -\frac{1}{\omega^{2m}} \exp\left(-\frac{2\pi \mathrm{i} \omega k'}{N}\right) + \sum_{k=1}^{N} \overline{C}_k \sum_{\beta \neq 0} \frac{\exp(2\pi \beta \mathrm{i}(k - k'))}{2\pi \beta^{2m}} = 0,$$

$$\text{for } k' = 1, 2, \ldots, N.$$

Taking into account (37), it follows immediately that $\nu = 0$.

In order to find unknown coefficients $C_k$, it is enough to solve the systems (38) and (39). The solution of this system, which we denote by $\mathring{C}_k$, $k = 1, 2, \ldots, N$ and $\mathring{\nu}$, is a stationary point for the function $\Lambda(C, \nu)$. From the theory of the Lagrange method, it follows that $\mathring{C}_k$ values are searching values of the coefficients for the quadrature formula, $k = 1, 2, \ldots, N$. They give the conditional minimum for the square of the norm $\|\ell\|$ provided that (8) holds.

By virtue of condition (37), we see that the extremal function $\psi_\ell(x)$, defined by (9), vanishes at the nodes of the quadrature formula (4), i.e., $\psi_\ell\left(\frac{2\pi k}{N}\right) = 0$. This proves Babuška's theorem.

Here, we assume that the systems (38) and (39) are solvable. Its solvability follows from the general theory of the Lagrange multipliers. However, as is shown in the calculations, the matrix of the systems (38) and (39) coincides with the matrix of the system considered in [10] in the construction of optimal cubature formulas in the Sobolev space $\widetilde{L}_2^{(m)}$ of periodic functions. In [10], uniqueness of the set of the optimal coefficients was also proved. Hence, it immediately follows that the systems (38) and (39) have a unique solution.

## 5. Optimal Coefficients of the Quadrature Formula (4)

In the present section, we prove Theorem 2. In order to prove the theorem, it is enough to solve the systems (38) and (39) with respect to coefficients $C_k$, $k = 1, 2, \ldots, N$. For a solution of this system, we search in the form

$$\mathring{C}_k = C(\omega, N, m) \exp\left(i\omega \frac{2\pi k}{N}\right), \tag{40}$$

where $C(\omega, N, m)$ is an unknown function. We are directly convinced that $\mathring{C}_k$ values, defined by Formula (40), $k = 1, 2, \ldots, N$, satisfy Equality (39). Substituting (40) into (38), we obtain

$$\frac{-1}{\omega^{2m}} \exp\left(\frac{2\pi i \omega k'}{N}\right) + C(\omega, N, m) \sum_{k=1}^{N} \exp\left(\frac{i\omega 2\pi k}{N}\right) \sum_{\beta \neq 0} \frac{\exp\left(\frac{2\pi i \beta(k-k')}{N}\right)}{2\pi \beta^{2m}} = 0. \tag{41}$$

We introduce the following notation:

$$z = C(\omega, N, m) \sum_{\beta \neq 0} \frac{\exp\left(\frac{-2\pi i \beta k'}{N}\right)}{2\pi \beta^{2m}} \sum_{k=1}^{N} \exp\left(\frac{2\pi i k}{N}(\omega + \beta)\right). \tag{42}$$

It is clear that

$$\sum_{k=1}^{N} \exp\left(\frac{2\pi i k}{N}(\omega + \beta)\right) = \begin{cases} 0, & \text{if } \frac{\omega+\beta}{N} \text{ is not integer,} \\ N, & \text{if } \frac{\omega+\beta}{N} \text{ is integer.} \end{cases}$$

Therefore, denoting $t = \frac{\omega+\beta}{N}$ and then $\beta = tN - \omega$, we rewrite (42) in the form

$$\begin{aligned}
z &= \frac{N}{2\pi} C(\omega, N, m) \exp\left(\frac{-2\pi i (tN - \omega)k'}{N}\right) \sum_{t=-\infty}^{\infty} \frac{1}{(tN - \omega)^{2m}} \\
&= \frac{N}{2\pi} C(\omega, N, m) \exp\left(\frac{2\pi i \omega k'}{N}\right) \sum_{t=-\infty}^{\infty} \frac{1}{(tN - \omega)^{2m}}.
\end{aligned} \tag{43}$$

From (41)–(43), it follows that

$$\frac{N}{2\pi} C(\omega, N, m) \exp\left(\frac{2\pi i \omega k'}{N}\right) \sum_{t=-\infty}^{\infty} \frac{1}{(tN - \omega)^{2m}} - \frac{1}{\omega^{2m}} \exp\left(\frac{2\pi i k' \omega}{N}\right) = 0,$$

$$k' = 1, 2, \ldots, N.$$

Hence, we have

$$\frac{1}{\omega^{2m}} \exp\left(\frac{2\pi i \omega k'}{N}\right) \left[\frac{N}{2\pi} C(\omega, N, m) \sum_{t=-\infty}^{\infty} \frac{1}{(1 - \frac{tN}{\omega})^{2m}} - 1\right] = 0,$$

$$k' = 1, 2, \ldots, N.$$

From the uniqueness of the solution for the systems (38) and (39), we obtain

$$C(\omega, N, m) = \frac{2\pi}{N} \left(\sum_{t=-\infty}^{\infty} \frac{1}{(1 - \frac{tN}{\omega})^{2m}}\right)^{-1}.$$

Hence, we directly obtain

$$C(\omega, N, m) = \frac{2\pi}{N} \left(\frac{N}{\omega}\right)^{2m} \left(\sum_{t=-\infty}^{\infty} \frac{1}{(t - \frac{\omega}{N})^{2m}}\right)^{-1}. \tag{44}$$

We now refer to calculations of

$$f(z) = \left(\sum_{t=-\infty}^{\infty} \frac{1}{(t-z)^{2m}}\right)^{-1}. \tag{45}$$

As is known [11],

$$\sum_{t=-\infty}^{\infty} \frac{1}{(z-t)^2} = \left(\frac{\pi}{\sin(\pi z)}\right)^2. \tag{46}$$

Taking the derivative of order $2m - 2$ with respect to $z$ from both sides of (46) and taking (45) into account, we obtain

$$\frac{1}{f(z)} = \frac{1}{(2m-1)!} \frac{d^{2m-2}}{dz^{2m-2}} \left(\frac{\pi}{\sin(\pi z)}\right)^2.$$

Now we calculate $\frac{d^{2m-2}}{dz^{2m-2}} \left(\frac{\pi}{\sin(\pi z)}\right)^2$. For this, we use Euler's known formula

$$\sin(\pi z) = \frac{\exp(2\pi i x) - 1}{2i \exp(\pi i z)}$$

Denoting $\lambda := \exp(2\pi i z)$, we have

$$\frac{d}{dz} = \frac{d\lambda}{dz} \frac{d}{d\lambda}, \frac{d}{dz} = 2\pi i \lambda \frac{d}{d\lambda},$$

$$\frac{d^{2m-2}}{dz^{2m-2}} = (2\pi i)^{2m-2} D^{2m-2},$$

where $D = \lambda \frac{d}{d\lambda}$, and $D^{2m-2} = \lambda \frac{d}{d\lambda} D^{2m-3}$.

Consequently,

$$\frac{1}{f(z)} = \frac{(2\pi i)^{2m}}{(2m-1)!} D^{2m-2} \frac{\lambda}{(1-\lambda)^2}. \tag{47}$$

Hence, as is known (see, for example, [18]), the Euler–Frobenius polynomial is defined by the formula

$$\lambda E_{2m-2}(\lambda) = (1-\lambda)^{2m} D^{2m-2} \frac{\lambda}{(1-\lambda)^2} \tag{48}$$

and

$$E_{2m-2}(\lambda) = \sum_{n=0}^{2m-2} a_n^{(2m-2)} \lambda^n,$$

where

$$a_n^{(2m-2)} = (n+1)^{2m-1} - \binom{2m}{1} n^{2m-1} + \binom{2m}{2}(n-1)^{2m-1} - \ldots (-1)^n \binom{2m}{n},$$

$$a_n^{(2m-2)} = a_{2m-2-n}^{(2m-2)}.$$

From (48), we have

$$D^{2m-2}\frac{\lambda}{(1-\lambda)^2} = \frac{\lambda E_{2m-2}(\lambda)}{(1-\lambda)^{2m}}. \tag{49}$$

By virtue of (49) and (47), Equality (45) takes the form

$$f(z) = \frac{(-1)^m (1-\lambda)^{2m}(2m-1)!}{(2\pi)^{2m}\lambda E_{2m-2}(\lambda)}.$$

Using the formula

$$\sin^2 \pi z = \frac{(1-\lambda)^2}{2i\lambda},$$

after some calculations, we obtain

$$f(z) = \frac{\sin^{2m}\pi z}{\pi^{2m}} \cdot \frac{\lambda^{m-1}(2m-1)!}{E_{2m-2}(\lambda)}.$$

By virtue of symmetry for the coefficients of Euler's polynomial, we have

$$
\begin{aligned}
f(z) &= \left(\frac{\sin \pi z}{\pi}\right)^{2m} \frac{\lambda^{m-1}(2m-1)!}{\sum\limits_{n=0}^{2m-2} a_n^{(2m-2)}\lambda^{2m-2-n}} \\
&= \left(\frac{\sin \pi z}{\pi}\right)^{2m} \frac{(2m-1)!}{\sum\limits_{n=0}^{m-2} a_n^{(2m-2)}\lambda^{m-1-n}} \\
&= \left(\frac{\sin \pi z}{\pi}\right)^{2m} \frac{(2m-1)!}{2\sum\limits_{n=0}^{m-2} a_n^{(2m-2)}(\lambda^{m-1-n}+\lambda^{n+1-m}) + a_{m-1}^{(2m-2)}} \\
&= \left(\frac{\sin \pi z}{\pi}\right)^{2m} \frac{(2m-1)!}{2\sum\limits_{n=0}^{m-2} a_n^{(2m-2)}\cos(2\pi(m-1-n))z + a_{m-1}^{(2m-2)}}.
\end{aligned}
\tag{50}
$$

Accordingly, from (50), (44), and (45), we obtain

$$C(\omega,N,m) = \frac{2\pi}{N}\left(\frac{\sin \frac{\pi\omega}{N}}{\frac{\pi\omega}{N}}\right)^{2m} \cdot \frac{(2m-1)!}{2\sum\limits_{n=0}^{m-2} a_n^{(2m-2)}\cos 2\pi(m-1-n)\frac{\omega}{N} + a_{m-1}^{(2m-2)}}. \tag{51}$$

From (40) and (51), the statement of Theorem 2 follows; i.e., the optimal coefficients of the quadrature formula (4) have the form

$$\mathring{C}_k = \frac{2\pi}{N}\left(\frac{\sin \frac{\pi\omega}{N}}{\frac{\pi\omega}{N}}\right)^{2m} \cdot \frac{(2m-1)!\exp\left(\frac{2\pi i\omega k}{N}\right)}{2\sum\limits_{n=0}^{m-2} a_n^{(2m-2)}\cos 2\pi(m-1-n)\frac{\omega}{N} + a_{m-1}^{(2m-2)}},$$

$$k = 1, 2, \ldots, N. \tag{52}$$

## 6. The Norm for the Error Functional of the Optimal Quadrature Formula

Here, we present the proof of Theorem 3. For this, we simplify the expression (36) as follows:

$$
\begin{aligned}
\|\mathring{\ell}|\widetilde{H}_2^{(m)^*}\|^2 &= \frac{2\pi}{\omega^{2m}} - \frac{1}{\omega^{2m}} \sum_{k'=1}^{N} \mathring{\overline{C}}_{k'} \exp\left(\frac{2\pi i k' \omega}{N}\right) - \frac{1}{\omega^{2m}} \sum_{k=1}^{N} \mathring{C}_k \exp\left(\frac{-2\pi i k \omega}{N}\right) \\
&\quad + \sum_{k=1}^{N} \sum_{k'=1}^{N} \mathring{C}_k \mathring{\overline{C}}_{k'} \sum_{\beta \neq 0} \frac{\exp\left(\frac{2\pi i \beta (k-k')}{N}\right)}{2\pi \beta^{2m}} = \frac{2\pi}{\omega^{2m}} - \sum_{k'=1}^{N} \mathring{\overline{C}}_{k'} \\
&\quad \times \left[ \frac{1}{\omega^{2m}} \exp\left(\frac{2\pi i k' \omega}{N}\right) - \sum_{k=1}^{N} \mathring{C}_k \sum_{\beta \neq 0} \frac{\exp\left(\frac{2\pi i \beta (k-k')}{N}\right)}{2\pi \beta^{2m}} \right] \\
&\quad - \frac{1}{\omega^{2m}} \sum_{k=1}^{N} \mathring{C}_k \exp\left(-\frac{2\pi i \omega k}{N}\right).
\end{aligned}
$$

Hence, taking (38) into account for the square of the norm, we obtain

$$
\|\mathring{\ell}|\widetilde{H}_2^{(m)^*}\|^2 = \frac{2\pi}{\omega^{2m}} - \frac{1}{\omega^{2m}} \sum_{k=1}^{N} \mathring{C}_k \exp\left(-\frac{2\pi i \omega k}{N}\right).
$$

Then, using the analytic expressions of the optimal coefficients, i.e., by the formula (52), we obtain

$$
\begin{aligned}
\|\mathring{\ell}|\widetilde{H}_2^{(m)^*}\|^2 &= \frac{2\pi}{\omega^{2m}} - \frac{2\pi}{N\omega^{2m}} \sum_{k=1}^{N} \left(\frac{\sin\frac{\pi\omega}{N}}{\frac{\pi\omega}{N}}\right)^{2m} \\
&\quad \times \frac{(2m-1)! \exp\left(\frac{2\pi i k \omega}{N}\right) \exp\left(-\frac{2\pi i k \omega}{N}\right)}{2 \sum_{n=0}^{m-2} a_n^{(2m-2)} \cos 2\pi(m-1-n)\frac{\omega}{N} + a_{m-1}^{(2m-2)}}.
\end{aligned}
$$

Provided that $\exp\left(\frac{2\pi i k \omega}{N}\right) \exp\left(-\frac{2\pi i k \omega}{N}\right) = 1$ and $\sum_{k=1}^{N} 1 = N$, for the square of the norm, we finally obtain

$$
\|\mathring{\ell}|\widetilde{H}_2^{(m)^*}\|^2 = \frac{2\pi}{\omega^{2m}} \left[ 1 - \left(\frac{\sin\frac{\pi\omega}{N}}{\frac{\pi\omega}{N}}\right)^{2m} \frac{(2m-1)!}{2 \sum_{n=0}^{m-2} a_n^{(2m-2)} \cos 2\pi(m-1-n)\frac{\omega}{N} + a_{m-1}^{(2m-2)}} \right].
$$

Theorem 3 is completely proved.

Now, we give the square of the norm of the error functional for the several first values of $m$ for $\omega < N$.

If $m = 1$, then

$$
\left\|\mathring{\ell}|\widetilde{H}_2^{(1)*}\right\|^2 = \frac{\pi}{3}\left(\frac{2\pi}{N}\right)^2 - \frac{\omega^2}{180}\left(\frac{2\pi}{N}\right)^4 + O\left(\left(\frac{2\pi}{N}\right)^6\right).
$$

Now let $m = 2$. It follows that

$$
\left\|\mathring{\ell}|\widetilde{H}_2^{(2)*}\right\|^2 = \frac{\pi}{\cos\frac{2\pi\omega}{N} + 2}\left[\frac{1}{5!}\left(\frac{2\pi}{N}\right)^4 + \frac{3\omega^2}{7!}\left(\frac{2\pi}{N}\right)^6\right] + O\left(\left(\frac{2\pi}{N}\right)^8\right).
$$

It is easy to see that, for $N \to \infty$ and $\omega < N$,

$$\left( \frac{\sin \frac{\pi \omega}{N}}{\frac{\pi \omega}{N}} \right)^{2m} \to 1 \text{ and}$$

$$2 \sum_{n=0}^{m-2} a_n^{(2m-2)} \cos\left( 2\pi (m-1-n) \frac{\omega}{N} \right) + a_{m-1}^{(2m-2)} \to 2 \sum_{n=0}^{m-2} a_n^{(2m-2)} + a_m^{(2m-2)} = (2m-1)!,$$

then $\|\ell\|^2 \to 0$.

In the case $\omega > N$ and $\omega \to \infty$, we have

$$\|\ell\|^2 \to \frac{2\pi}{\omega^{2m}}.$$

From the last results, we can conclude that, for the functions from the space $\widetilde{H}_2^m$, the rate of convergence of the constructed optimal quadrature formula is $O(h^m)$ for $|\omega| < N$ and $O(|\omega|^{-m})$ for $|\omega| \geq N$.

## 7. Numerical Results

In this section, we present numerical results of comparison for absolute errors of the optimal quadrature formula of the form (4) with the $\exp(i\omega x)$ weight in the case $m = 2$ and the Midpoint formula. We note that the rate of convergence for both of these formulas is $O(h^2)$. Here we use the Maple to obtain numerical results.

As an example, we consider calculation of the integral

$$I = \int_0^{2\pi} \exp(i\omega x)\varphi(x)dx \tag{53}$$

with $\varphi(x) = \frac{e^{1-x/(2\pi)} + e^{x/(2\pi)}}{2(1-e)}$. Since for the functions $\varphi$ we have the relation $\varphi(0) = \varphi(2\pi)$, we can consider it as a periodic function on the interval $[0, 2\pi]$. For convenience, we denote the integrand as $f(x)$; i.e., $f(x) = \exp(i\omega x)\frac{e^{1-x/(2\pi)} + e^{x/(2\pi)}}{2(1-e)}$.

We approximately calculate the integral $I$ by the Midpoint rule. Then approximate value for the integral (53) is then calculated as follows using the Midpoint rule

$$A_{\mathrm{mid}} = \sum_{k=1}^{N} f\left( \frac{x_k + x_{k-1}}{2} \right)(x_k - x_{k-1}), \tag{54}$$

where $x_k = kh$, $k = 0, 1, \dots, N$ and $h = 2\pi/N$.

Hence, for the function $f(x) = \exp(i\omega x)\frac{e^{1-x/(2\pi)} + e^{x/(2\pi)}}{2(1-e)}$, the error of the Midpoint rule (54) is

$$I - A_{\mathrm{mid}} = \int_0^{2\pi} f(x)dx - \sum_{k=1}^{N} f\left( \frac{x_k + x_{k-1}}{2} \right)(x_k - x_{k-1}). \tag{55}$$

In Table 1, we give the absolute values for the real part of the error (55) of the Midpoint rule for $N = 1, 10, 100, 1000$ and $\omega = 1, 10, 100, 1000$.

**Table 1.** The absolute values for the real part of the error (55) of the Midpoint rule for $N = 1$, 10, 100, 1000 and $\omega = 1$, 10, 100, 1000.

|  | $\omega = 1$ | $\omega = 10$ | $\omega = 100$ | $\omega = 1000$ |
|---|---|---|---|---|
|  | $\lvert I - A_{\mathbf{mid}} \rvert$ | $\lvert I - A_{\mathbf{mid}} \rvert$ | $\lvert I - A_{\mathbf{mid}} \rvert$ | $\lvert I - A_{\mathbf{mid}} \rvert$ |
| $N = 1$ | 6.184049 | 6.027234 | 6.028810 | 6.028825 |
| $N = 10$ | 2.709174(−3) | 6.282159 | 6.280552 | 6.280552 |
| $N = 100$ | 2.618891(−5) | 2.710011(−5) | 6.283175 | 6.283159 |
| $N = 1000$ | 2.618003(−7) | 2.618898(−7) | 2.710020(−7) | 6.283185 |

It can be seen from the results given in Table 1 that the Midpoint rule converges for $N > \omega$. In Figure 1, the process of this convergence is graphically shown when $\omega = 1$ and $N = 1, 10, 100, 1000$ for the real part of the function $f(x)$.

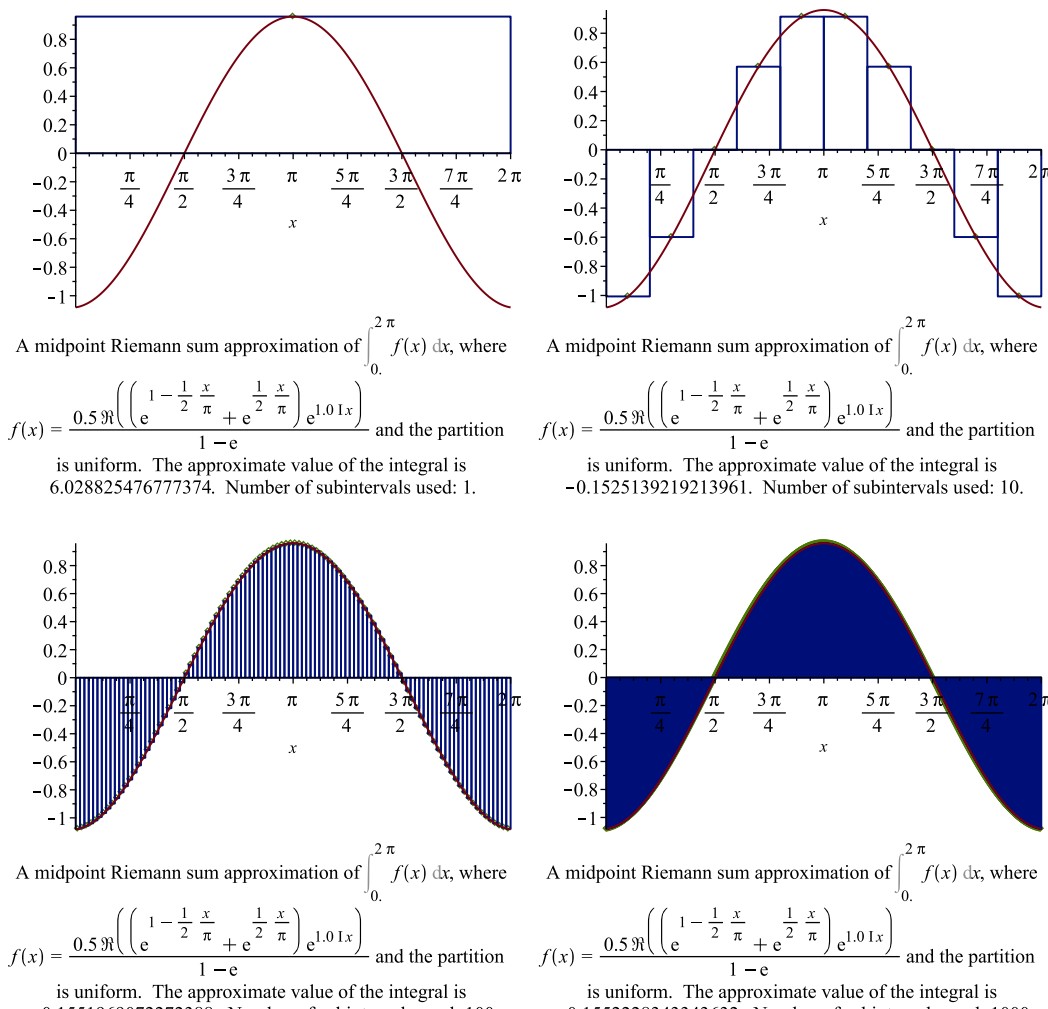

**Figure 1.** The process of convergence for the Midpoint rule when $\omega = 1$ and $N = 1, 10, 100, 1000$ for the real part of the function $f(x)$.

In Figure 2 the graphs of numerical calculation of the integral (53) by the Midpoint rule for the case $\omega = 1, 10, 100, 1000$ and $N = 1$ are given. Here, we can see that the Midpoint rule does not converge when $\omega \geq N$ for the function $f(x)$.

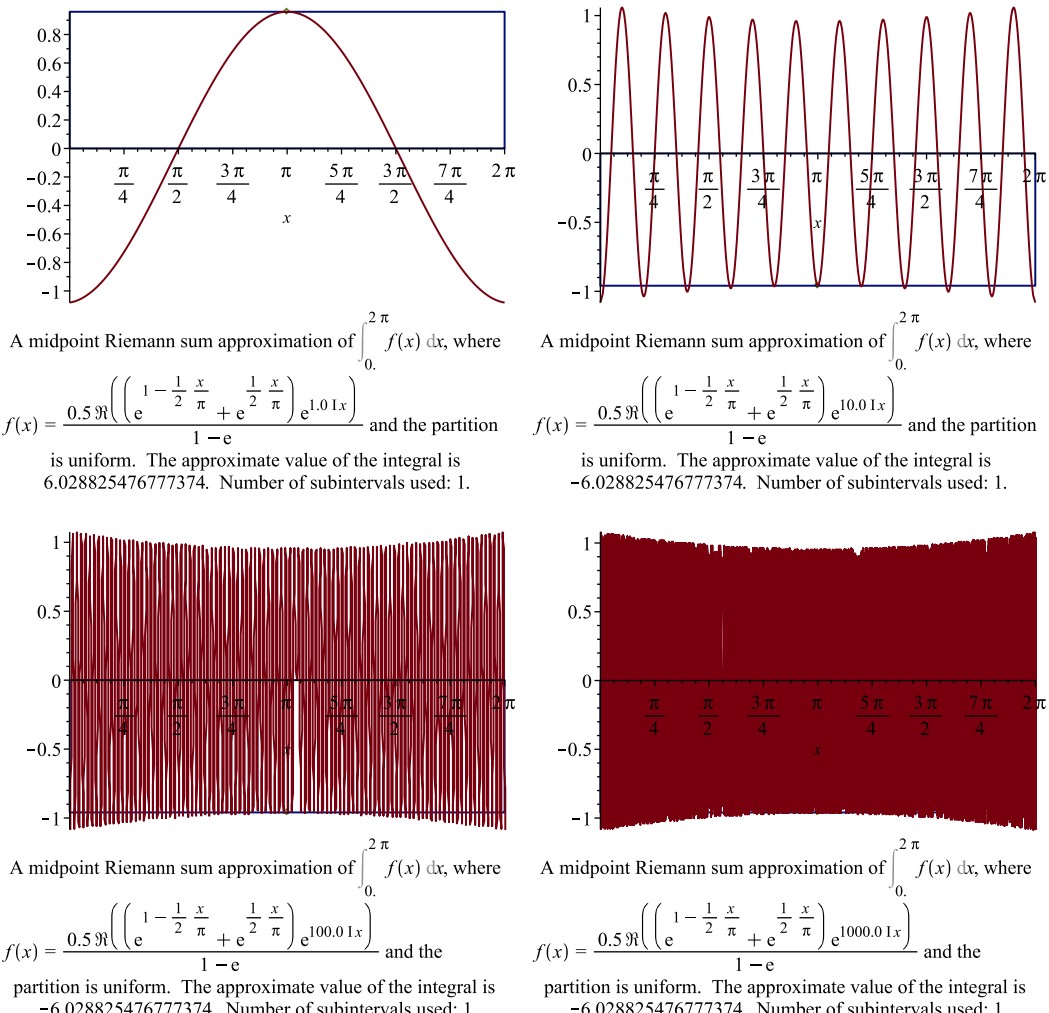

**Figure 2.** The Midpoint rule does not converge when $\omega \geq N$ for the function $f(x)$.

Now, we approximate the above integral (53) using the optimal quadrature formula of the form (4) with $\exp(\mathrm{i}\omega x)$ weight in the case $m = 2$. Then, taking into account (10), we have the approximate equality

$$\int_0^{2\pi} \exp(\mathrm{i}\omega x)\varphi(x)\,dx \cong \sum_{k=1}^{N} \mathring{C}_k \varphi\left(\frac{2\pi k}{N}\right), \tag{56}$$

for $\varphi(x) = \frac{e^{1-x/(2\pi)}+e^{x/(2\pi)}}{2(1-e)}$ with the optimal coefficients

$$\mathring{C}_k = \frac{2\pi}{N}\left(\frac{\sin\left(\frac{\pi\omega}{N}\right)}{\frac{\pi\omega}{N}}\right)^4 \frac{3\exp\left(\frac{2\pi \mathrm{i}\omega k}{N}\right)}{\cos\left(\frac{2\pi\omega}{N}\right)+2}, \quad k = 1, 2, \ldots, N.$$

The approximate value for the integral (53) is calculated as follows, using the optimal quadrature formula:

$$A_{\mathrm{opt}} = \sum_{k=1}^{N} \mathring{C}_k \frac{e^{1-\frac{k}{N}}+e^{\frac{k}{N}}}{2(1-e)}.$$

Hence, for the function $\varphi(x) = \frac{e^{1-x/(2\pi)}+e^{x/(2\pi)}}{2(1-e)}$, the error of the optimal quadrature formula (56) is

$$I - A_{\text{opt}} = \int_0^{2\pi} \exp(\mathrm{i}\omega x)\frac{e^{1-x/(2\pi)}+e^{x/(2\pi)}}{2(1-e)}dx - \sum_{k=1}^{N} \mathring{C}_k \frac{e^{1-\frac{k}{N}}+e^{\frac{k}{N}}}{2(1-e)}. \tag{57}$$

Thus, the numerical results of Table 2 show convergence of the optimal quadrature formula (56) for $N \geq \omega$ and $N < \omega$.

**Table 2.** The absolute values for the real part of the error (57) of the optimal quadrature formula (56) for $N = 1, 10, 100, 1000$ and $\omega = 1, 10, 100, 1000$.

|  | $\omega = 1$ | $\omega = 10$ | $\omega = 100$ | $\omega = 1000$ |
|---|---|---|---|---|
|  | $|I - A_{\text{opt}}|$ | $|I - A_{\text{opt}}|$ | $|I - A_{\text{opt}}|$ | $|I - A_{\text{opt}}|$ |
| $N = 1$ | $1.552231(-1)$ | $1.591146(-3)$ | $1.591545(-5)$ | $1.591549(-7)$ |
| $N = 10$ | $5.301897(-3)$ | $1.591146(-3)$ | $1.591545(-5)$ | $1.591549(-7)$ |
| $N = 100$ | $5.236676(-5)$ | $5.301920(-5)$ | $1.591545(-5)$ | $1.591549(-7)$ |
| $N = 1000$ | $5.235995(-7)$ | $5.236677(-7)$ | $5.301920(-7)$ | $1.591549(-7)$ |

## 8. Conclusions

We obtained optimal quadrature formulas for the approximate calculation of the Fourier coefficients $\int_0^{2\pi} e^{\mathrm{i}\omega x}\varphi(x)dx$ in Sobolev space $\tilde{H}_2^m$. Firstly, in order to obtain an upper bound for the error of the quadrature formula, the norm of the error functional was calculated. Using the extremal function of the considered quadrature formula, we calculated the norm of the error functional. We found the explicit forms of the coefficients for the optimal quadrature formula, and they provide the minimum value to the norm of the error functional. Finally, we calculated the norm of the error functional of the optimal quadrature formulas. We show that, for the functions from the space $\tilde{H}_2^m$, the rate of convergence of the constructed optimal quadrature formula is $O(h^m)$ for $|\omega| < N$ and $O(|\omega|^{-m})$ for $|\omega| \geq N$. Finally, we presented numerical results of comparison for absolute errors of the constructed optimal quadrature formula in the case $m = 2$ and the Midpoint formula, showing the advantage of the optimal quadrature formulas.

**Author Contributions:** The problem of the manuscript was stated by K.S. and A,H.; proofs of the main theoretical results were obtained by K.S., A.H. and B.A.; the numerical results were obtained by B.A.. All authors have read and agreed to the published version of the manuscript.

**Funding:** This research received no external funding.

**Institutional Review Board Statement:** The study was conducted in accordance with the Scientific program of the Computational Mathematics Laboratory of V. I. Romanovskiy Institute of Mathematics and discussed in the Scientific seminar of the Laboratiry (protocol code 7, 8 July 2022).

**Informed Consent Statement:** Not applicable

**Data Availability Statement:** Not applicable.

**Acknowledgments:** We are very thankful to the reviewers for valuable comments and remarks that have improved the quality of the paper.

**Conflicts of Interest:** The authors declare that they have no competing interests.

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
