# Peer review of "On an Optimal Quadrature Formula in a Hilbert Space of Periodic Functions"

_algorithms, doi:10.3390/a15100344_

Round 1

Reviewer 1 Report

see the attached pdf file

Author Response

Response to Reviewer 1 Comments

Point 1: My main objection is that the paper in its present form is purely theoretical and therefore may not fully fit into the scope of Algorithms. Therefore, I suggest including in the paper a numerical simulation which will show in practice that the new quadrature formula leads to a better or faster approximation in comparison to the classical methods. Final estimates of Section 6 suggest that such an effect is expected.

Response 1:

Yes, we agree with the Reviewer.

At the end of the paper we added “Numerical results” section. In that section we compared the optimal quadrature formula in the case m=2 and the Midpoint rule. We note that the rate of convergence for both of these formulas is O(h^2). It is easy to see the advantage of the optimal quadrature formulas.

In addition, we have added the section:

”Acknowledgments”

We are very thankful to the reviewers for valuable comments and remarks which have improved the quality of the paper.

September 18, 2022                                                                                                           Authors

Reviewer 2 Report

Comments and suggestions are attached.

Author Response

Response to Reviewer 2 Comments

Point 1: The general topic is relevant and of interest to the scientific community. Furthermore, it is appropriate for the journal under consideration. The authors prove a deep knowledge of the area and the results are interesting, scientifically sound with appropriate supplementary materials and constitute a significant contribution to the field. However, considering the proper use of the English language, there is quite some need for action. With well over 50 remarks in this direction a complete

list of all changes needed is not feasible, especially with the amount of gaps in the line numbering provided by the journal, which make a proper assignment of comments unnecessarily difficult. Hence, a commented version of the article is at tached to the report. These comments need to be addressed before the manuscript can be accepted for publication.

Accordingly, I recommend this paper for publication in Algorithms, subject to the changes outlined in the document attached.

Response 1:

We have done all changes outlined in the document attached by the Reviewer.

In addition, we have added the section:

”Acknowledgments”

We are very thankful to the reviewers for valuable comments and remarks which have improved the quality of the paper.

September 18, 2022                                                                                                                                                            Authors

Reviewer 3 Report

The English is fine, except some minor editing.   I think that the paper is scientific sound. My profile is that of an economist, and it is bette to have someone checking all the results.   Finally, there is a good degree of novelty, and I also recommended having some applications included.  

Author Response

Response to Reviewer 3 Comments

Point 1:  The english is fine, except some minor editiong. I think that the paper is scientific sound. My profile is that of an economist, and it is better to have someone cheking all the results. Finally, there is a good degree of novelty and I also recommendet having some applications included.

Response 1: English is improved. At the end of the paper we have added “Numerical results” section, where it can be seen the advantage of the optimal quadrature formulas.

In addition, we have added the section:

”Acknowledgments”

We are very thankful to the reviewers for valuable comments and remarks which have improved the quality of the paper.

September 18, 2022                                                                                                                                                            Authors
